# SongEval: A Benchmark Dataset for Song Aesthetics Evaluation

## Abstract

Aesthetics serve as an implicit and important criterion in song generation tasks that reflect human perception beyond objective metrics. However, evaluating the aesthetics of generated songs remains a fundamental challenge, as the appreciation of music is highly subjective. Existing evaluation metrics, such as embedding-based distances, are limited in reflecting the subjective and perceptual aspects that define musical appeal. To address this issue, we introduce SongEval, the first open-source, large-scale benchmark dataset for evaluating the aesthetics of full-length songs. SongEval includes over 2,399 songs in full length, summing up to more than 140 hours, with aesthetic ratings from 16 professional annotators with musical backgrounds. Each song is evaluated across five key dimensions: overall coherence, memorability, naturalness of vocal breathing and phrasing, clarity of song structure, and overall musicality. The dataset covers both English and Chinese songs, spanning nine mainstream genres. Moreover, to assess the effectiveness of song aesthetic evaluation, we conduct experiments using SongEval to predict aesthetic scores and demonstrate better performance than existing objective evaluation metrics in predicting human-perceived musical quality. We provide the dataset and toolkit for song aesthetic evaluation at `https://anonymous.4open.science/r/SongEval_anonymous-7505`

## 1 Introduction

Song generation lies at the intersection of structured pattern learning and human aesthetics. With the advancement of deep learning-based generative models, current approaches can now compose melodies, harmonies, and full musical pieces that closely resemble human-created songs Dhariwal et al. (2020); Li et al. (2024); Lei et al. (2024); Liu et al. (2025b); Yang et al. (2025). This progress has enabled a wide range of applications, including personalized music and song generation for games, film scoring, music education tools, and therapeutic settings. As a universal medium of expression and communication, song generation increasingly aims to produce songs that are both aesthetically pleasing and emotionally resonant. However, evaluating the aesthetic quality of generated songs remains an open and underexplored challenge, primarily due to the subjective and multi-dimensional nature of musical aesthetics.

A typical song consists of two main components: the singing voice and the instrumental accompaniment. As shown in Figure 1, these components work together to convey the musical message, where vocals deliver melody, lyrics, and emotional expression and the accompaniment provides rhythmic and stylistic support. Most previous studies only focus on single-component generation, such as singing voice synthesis Liu et al. (2022); Zhang et al. (2022); Ye et al. (2023); Zhang et al. (2024); Hwang et al. (2025) or text-to-music generation Agostinelli et al. (2023); Huang et al. (2023a); Liu et al. (2023); Huang et al. (2023b); Majumder et al. (2024). As a result, there remains a gap in generating full-length songs that seamlessly integrate both vocals and accompaniment in a coherent and aesthetically pleasing way. Recently, some studies Yuan et al. (2025); Ning et al. (2025); Lam et al. (2025a); Bai et al. (2024) have scaled up model parameters and training corpora to directly generate full-length songs that combine vocals and accompaniment with greater coherence and aesthetic quality. These approaches have attracted significant interest from both industry and academia.

A critical challenge in song generation is evaluating the quality of the generated song, particularly given that songs are deeply rooted in aesthetic experience. While objective metrics such as mel-

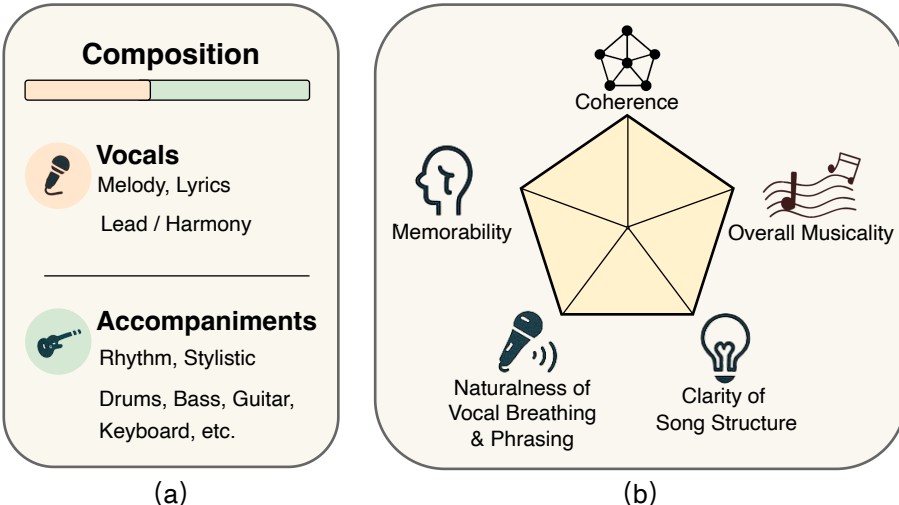

Figure 1: Aesthetic evaluation dimensions and structural components of a song. (a) Structural components of a song. (b) Five aesthetic dimensions used in SongEval for full-length song evaluation.

spectrogram distance, pitch accuracy, and embedding-based similarity offer insights into signal-level or structural fidelity, they fall short of capturing the subjective and multifaceted nature of musical aesthetics. These low-level distance measures do not account for how human listeners perceive qualities such as emotional expressiveness, coherence between vocals and accompaniment, or overall musicality. Consequently, there remains a significant gap in the current evaluation pipeline of song generation, limiting the development and comparison of song generation models designed to produce aesthetically pleasing music.

To facilitate aesthetic evaluation in song generation, we introduce SongEval, a large-scale, open-source dataset containing over 140 hours of professionally annotated songs with human aesthetic ratings. The annotations are provided by 16 expert raters with formal musical education, ensuring high reliability and perceptual consistency rooted in professional musical understanding. Each song in the dataset is evaluated across five complementary aesthetic dimensions: overall coherence, memorability, naturalness of vocal breathing and phrasing, clarity of song structure, and overall musicality. These dimensions are carefully selected to reflect the preferences and evaluative standards of professionally trained musicians, aligning the metric with academic and industry-level expectations. It is important to note that our definition of aesthetic quality is not intended to represent personalized taste. Rather, it approximates the consensus of expert musicians, providing a reliable, authoritative evaluation dataset for assessing song generative models. While no single metric can fully capture the complexity of musical aesthetics due to its inherently subjective nature, our goal is not to define a perfect metric but to establish one that is more explainable and professionally aligned than previous alternatives. By providing high-quality, multi-dimensional aesthetic annotations at scale, SongEval establishes a new paradigm for benchmarking generative models based on professionally informed musical evaluation, offering a valuable resource for improving and comparing song generation systems.

## 2 RELATED WORK

Recent advancements in generative models have led to remarkable improvements in the quality of synthesized audio, including speech, music, and general sound. High-fidelity generation has become increasingly achievable, yet evaluating the perceptual quality of these outputs remains an open and urgent challenge. Particularly in the context of human perception, objective signal-based metrics often fail to reflect how listeners actually experience generated audio. In the speech domain, this gap has been addressed through the development of subjective evaluation datasets, such as those providing human-annotated Mean Opinion Scores (MOS) Reddy et al. (2022); Lorenzo-Trueba et al. (2018); Zhao et al. (2020), which are now widely adopted to train prediction toolkits to benchmark speech synthesis systems Saeki et al. (2022); Lo et al. (2019).

Table 1: Comparison between proposed SongEval and other similar subjective evaluation datasets.

|  |  | MusicEval | AES-Natural | SongEval |
|---|---|---|---|---|
| Language |  | - | EN | EN & ZH |
| Total Hours |  | 16.67 | 29.44 | 140.32 |
| Utt. Average Duration (min) |  | 0.36 | 1.77 | 3.51 |
| Components |  | Accompaniments only | Accompaniments +Vocal | Accompaniments +Vocal |
| Annotation Aspects | Musicality | ✓ | ✓ | ✓ |
|  | Clarity | ✗ | ✓ | ✓ |
|  | Naturalness | ✗ | ✓ | ✓ |
|  | Memorability | ✗ | ✗ | ✓ |
|  | Coherence | ✗ | ✗ | ✓ |

In contrast, subjective evaluation datasets for music and audio remain limited. MusicEval Liu et al. (2025a) is one of the few efforts in this area, focusing solely on accompaniment generation and offering approximately 16 hours of annotated data. However, it provides ratings only for musical impression and alignment with the description prompt, lacking fine-grained aesthetic dimensions and excluding full-length songs with vocals. AES-Natural Tjandra et al. (2025b) offers a broader scope across speech, audio, and music, with a total of 29 hours of data. While it includes some music clips, the segments are short and do not represent full-song structures. Additionally, the dataset evaluates generation quality along only three basic dimensions, offering limited insight into the nuanced perception of musical aesthetics. These limitations highlight the need for a comprehensive dataset that supports multi-dimensional aesthetic evaluation of full-length songs, which is the goal of SongEval. The detailed comparisons between our proposed SongEval and other similar music evaluation datasets are shown in Table 1.

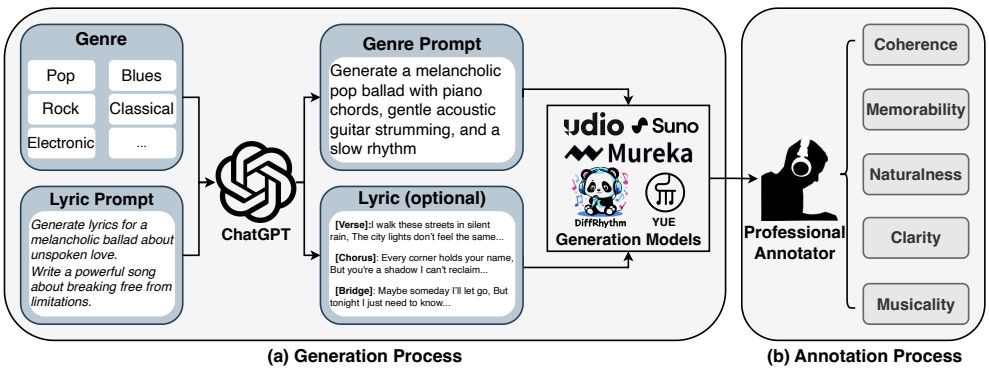

Figure 2: The data collection pipeline of SongEval. Lyrics are an optional input, as some commercial systems can generate songs using only a genre prompt.

## 3 SONGEVAL DATASET

In this section, we introduce SongEval, a large-scale benchmark dataset comprising full-length songs with expert-annotated aesthetic ratings. We begin by describing the data collection pipeline, including the generation of input conditions and the production of final full-length songs. Next, we detail the annotation protocol, including the five key aesthetic dimensions used to evaluate each song. Finally, we provide statistical insights into the dataset.

### 3.1 DATA COLLECTION

The construction of the SongEval dataset involves two key stages: (1) the generation of lyrics and genre-aligned prompts, and (2) the synthesis of full-length songs guided by these inputs, as shown

in Figure 2(a). In the first stage, we utilize ChatGPT Achiam et al. (2023) to generate lyrics and corresponding prompts conditioned on various musical genres. The genre includes nine categories: blues, pop, rock, classical, jazz, electronic, hip-hop, world music, and country, while each prompt describes the intended mood, style, and instrumentation of the desired song. The paired lyrics reflect thematic content and linguistic patterns commonly associated with the target genre and contain both English and Mandarin. This approach ensures genre diversity and stylistic richness across the dataset. The curated lyric-prompt pairs serve as the input for the second stage of music generation.

Table 2: The detailed information of SongEval. The gender duration ratio computes the total duration between male singers and female singers.

| Genre | Language | Duration (Hours) | Samples | Gender Duration Ratio |
|---|---|---|---|---|
| Pop | ZH | 15.74 | 284 | 52% / 48% |
| | EN | 11.28 | 175 | 57% / 43% |
| Rock | ZH | 5.29 | 91 | 61% / 39% |
| | EN | 14.33 | 233 | 64% / 36% |
| Electronic | ZH | 6.78 | 123 | 55% / 45% |
| | EN | 6.96 | 126 | 50% / 50% |
| Blues | ZH | 3.62 | 60 | 66% / 34% |
| | EN | 8.70 | 135 | 74% / 26% |
| World Music | ZH | 5.21 | 103 | 59% / 41% |
| | EN | 6.34 | 125 | 55% / 45% |
| Hip-hop/Rap | ZH | 4.35 | 83 | 65% / 35% |
| | EN | 3.31 | 62 | 79% / 21% |
| Country | ZH | 4.19 | 84 | 61% / 39% |
| | EN | 4.74 | 71 | 53% / 47% |
| Jazz | ZH | 4.13 | 69 | 50% / 50% |
| | EN | 4.09 | 64 | 60% / 40% |
| Classical | ZH | 3.71 | 62 | 43% / 57% |
| | EN | 2.77 | 43 | 32% / 68% |
| Others | ZH | 9.58 | 134 | 75% / 25% |
| | EN | 15.21 | 272 | 56% / 44% |
| **All** | - | **140.32** | **2399** | **60% / 40%** |

In the second stage, we use the generated lyric and genre prompt pairs as inputs to generate full-length songs using five mainstream song generation models Yuan et al. (2025); Ning et al. (2025); Lam et al. (2025b); Suno (2024); Udio (2024). These models are selected to cover a broad range of generation strategies and stylistic capacities, ensuring diversity in both vocal and instrumental characteristics. Each model takes the prompt as conditioning information and uses the associated lyrics as semantic guidance for vocal melody and lyrical content. The characteristic details of generated songs from different systems are provided in Appendix A.2. Moreover, since some commercial systems can generate songs using only a genre prompt, we adopt both genre-only and lyric–genre pair generation strategies to ensure a comprehensive and diverse collection. The detailed information about SongEval is shown in Table 2. After generation, we apply the vocal range as a metric to identify and remove low-quality outputs Yuan et al. (2025), details can be found in Appendix A.3.

### 3.2 AESTHETIC ANNOTATION

To enable fine-grained, multi-dimensional evaluation of generated songs, each sample in the SongEval is annotated across five aesthetic dimensions. These dimensions are carefully designed to capture key perceptual qualities that professional annotators consider when evaluating musical aesthetics. Each dimension is rated on a five-point scale, with 1 indicating the lowest quality and 5 indicating the highest. Definitions for each point on the 1-5 scale are provided in Appendix A.4. We provide

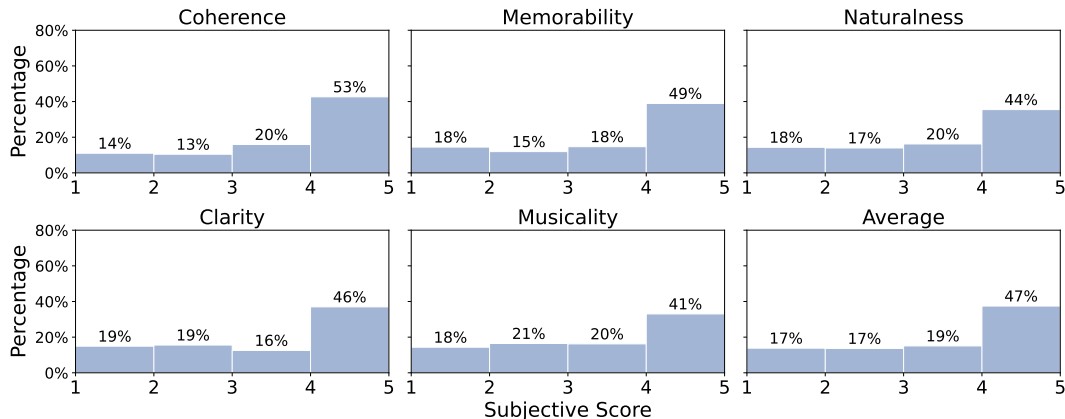

Figure 3: Distribution of overall subjective scores over five evaluation dimensions.

demos with representative examples for each aesthetic dimension [1]. The five aesthetic dimensions are defined as follows:

- Overall coherence: This dimension evaluates the musical and emotional continuity across different sections of the song, including intro, verse, chorus, and outro. High scores reflect smooth transitions, consistent dynamics, and a unified emotional tone throughout the piece.

- Memorability: This refers to the presence of distinctive musical features, such as a catchy melody, rhythmic motif, or lyrical hook, that make the song easy to remember after a single listen. Highly memorable songs typically exhibit a strong, repeatable musical identity.

- Naturalness of vocal breathing and phrasing: This dimension evaluates the phrasing quality and breath control in the vocal performance. It considers whether the phrasing aligns well with semantic breaks and rhythmic cues, and whether the breathing patterns support a fluent, natural delivery without disrupting the singing flow.

- Clarity of song structure: This dimension measures how clearly the song is structured into recognizable sections (e.g., verse, chorus, bridge), as well as the logic and coherence of the structural design. Both conventional structures and well-executed novel structures can achieve high scores, provided the segmentation is clear and musically meaningful.

- Overall musicality: This is an overall evaluation of listening enjoyment, considering factors such as melody, harmony, instrumentation, and the integration between vocals and accompaniment. It reflects the general aesthetic satisfaction a listener derives from the song.

Each song in the dataset is independently rated by four annotators with formal musical training, ensuring high-quality and reliable aesthetic annotations. Detailed information about the annotation process is provided in Appendix A.5. These ratings form the foundation for benchmarking generative models based on human musical perception. The score distribution for each aesthetic dimension across the five-point scale is illustrated in Figure 3.

### 3.3 DATASET STATISTICS

The final SongEval dataset consists of 2,399 full-length songs, totaling approximately 140 hours of audio. In terms of duration, most Chinese songs range from 2 to 6 minutes, while English songs follow a similar pattern, with some extending up to 8 minutes. This broad range captures both short-form pieces and structurally rich long-form content. The dataset includes songs in both English and Chinese, reflecting diverse linguistic and cultural backgrounds. To ensure stylistic variety, the collection also spans nine widely common song genres.

We also provide a breakdown of the dataset based on the song generation models used in the synthesis process. As shown in Figure 4, the dataset includes outputs from five mainstream song generation models, with DiffRhythm Ning et al. (2025) contributing the largest number of samples. This

---

[1] https://anonymous.4open.science/w/SongEval_anonymous-AB26/

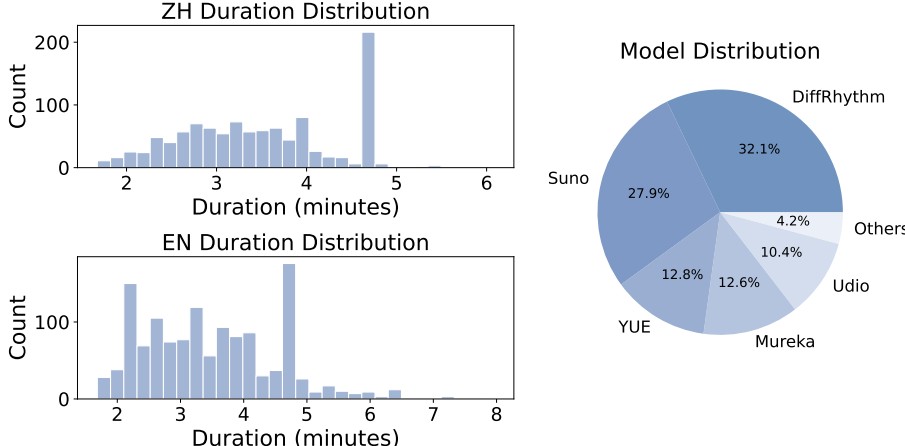

Figure 4: Duration distribution across different languages and generation models. Since the songs generated by DiffRhythm have a fixed duration of 285 seconds, a noticeable concentration of songs around the four-minute mark in the distribution.

distribution ensures model-level diversity and supports cross-model evaluation in downstream tasks. To facilitate robustness testing, we include a small subset of problematic cases, such as incomplete, off-pitch, or speech-like samples, and non-copyrighted real songs. These are grouped under a separate "Other" category and serve as valuable references for validation and double-checking.

## 4 EXPERIMENTAL SETUP

### 4.1 DATASETS

To evaluate the quality and versatility of SongEval, we conduct a comprehensive experiment on the task of song aesthetic evaluation. Specifically, we use a subset of SongEval that includes the aesthetic annotations described in Section 3.2. This dataset serves as a reliable and diverse training source for modeling subjective perceptions of musical quality. We randomly select 2,199 songs for training and reserve 200 songs as the evaluation set. Additionally, we include 50 non-copyrighted real songs to evaluation set to further assess the model's generalization ability. The evaluation set is used to test performance on unseen musical content.

### 4.2 MODELS

To evaluate the effectiveness and generalizability of SongEval, we conduct experiments using four representative approaches adapted from published studies in related fields. These systems are selected for their diverse architectural foundations and strong performance in speech and audio quality evaluation tasks, making them well-suited for adaptation to the music domain. All models are trained on the SongEval training set using eight NVIDIA A6000 GPUs, each with 48 GB of memory. Implementation details are provided below.

**MosNet-based**: A widely used baseline for non-intrusive speech quality assessment. MOSNet Lo et al. (2019) consists of convolutional and BLSTM layers followed by dense layers for subjective score prediction. We adapt it to process full-length songs and regress aesthetic ratings instead of speech MOS.

**LDNet-based**: LDNet Huang et al. (2022b) is a MOS prediction framework to predict listener-dependent scores, which combines two inference methods that provide stable results and efficient computation. Its efficiency and compactness make it a strong baseline for modeling speech perception.

**SSL-based**: A model that leverages self-supervised learning (SSL) audio representations, followed by a regression head to predict quality scores. We adopt the version originally designed by Cooper

et al. (2022) in speech synthesis tasks, adapting it to our five-dimensional aesthetic scoring framework and replacing the original SSL model with MuQ Zhu et al. (2025) [2].

**UTMOS-based**: Based on the UTokyo-SaruLab MOS prediction framework Saeki et al. (2022), this model is based on the ensemble learning of strong and weak learners and obtains the highest score on several metrics for both the main and out-of-distribution tracks on VoiceMOS 2022 Challenge Huang et al. (2022a).

Table 3: Multi-dimensional comparison results of different song aesthetic prediction systems between utterance-level and system-level.

| System | | Utterance-level | | | | System-level | | | |
|---|---|---|---|---|---|---|---|---|---|
| | | MSE↓ | LCC↑ | SRCC↑ | KATU↑ | MSE↓ | LCC↑ | SRCC↑ | KATU↑ |
| Coherence | MOSNet-based | 0.339 | 0.882 | 0.854 | 0.679 | 0.187 | 0.923 | 0.904 | 0.751 |
| | LDNet-based | 0.421 | 0.882 | 0.860 | 0.684 | 0.238 | 0.948 | 0.934 | 0.793 |
| | SSL-based | 0.237 | 0.900 | 0.882 | 0.719 | 0.088 | 0.959 | **0.962** | **0.860** |
| | UTMOS-based | **0.195** | **0.917** | **0.898** | **0.741** | **0.073** | **0.962** | 0.954 | 0.844 |
| Memorability | MOSNet-based | 0.360 | 0.874 | 0.851 | 0.672 | 0.206 | 0.919 | 0.889 | 0.727 |
| | LDNet-based | 0.547 | 0.867 | 0.846 | 0.671 | 0.340 | 0.936 | 0.920 | 0.776 |
| | SSL-based | 0.276 | 0.897 | 0.891 | 0.723 | 0.104 | 0.951 | 0.945 | 0.810 |
| | UTMOS-based | **0.241** | **0.910** | **0.901** | **0.739** | **0.096** | **0.955** | **0.958** | **0.849** |
| Naturalness | MOSNet-based | 0.406 | 0.843 | 0.818 | 0.634 | 0.203 | 0.923 | 0.901 | 0.740 |
| | LDNet-based | 0.449 | 0.867 | 0.855 | 0.688 | 0.247 | 0.924 | 0.911 | 0.763 |
| | SSL-based | 0.243 | 0.896 | 0.885 | 0.718 | **0.079** | 0.955 | **0.942** | **0.820** |
| | UTMOS-based | **0.219** | **0.909** | **0.896** | **0.734** | 0.081 | **0.957** | 0.941 | 0.809 |
| Clarity | MOSNet-based | 0.354 | 0.876 | 0.855 | 0.675 | 0.186 | 0.925 | 0.919 | 0.757 |
| | LDNet-based | 0.450 | 0.862 | 0.853 | 0.677 | 0.249 | 0.925 | 0.916 | 0.773 |
| | SSL-based | 0.235 | 0.903 | 0.889 | 0.720 | **0.085** | **0.952** | **0.951** | **0.824** |
| | UTMOS-based | **0.221** | **0.908** | **0.894** | **0.728** | 0.091 | 0.951 | 0.939 | 0.804 |
| Musicality | MOSNet-based | 0.337 | 0.877 | 0.854 | 0.677 | 0.168 | 0.934 | 0.928 | 0.784 |
| | LDNet-based | 0.466 | 0.881 | 0.861 | 0.689 | 0.262 | 0.944 | 0.927 | 0.779 |
| | SSL-based | 0.220 | 0.908 | 0.893 | 0.733 | **0.066** | 0.965 | **0.970** | **0.864** |
| | UTMOS-based | **0.203** | **0.916** | **0.901** | **0.745** | 0.072 | **0.966** | 0.969 | 0.859 |

## 4.3 EVALUATION METRICS

To quantitatively evaluate the performance of aesthetic prediction models trained on the SongEval dataset, we adopt four widely used metrics that assess the alignment between model-predicted scores and human-annotated scores across the five aesthetic dimensions. These metrics capture both absolute prediction accuracy and relative ranking quality: **Mean Squared Error (MSE)**: MSE measures the average squared difference between predicted scores and ground truth annotations. Lower MSE values indicate more accurate absolute predictions across samples. **Linear Correlation Coefficient (LCC)**: LCC Sedgwick (2012) quantifies the linear relationship between predicted and ground truth scores, reflecting how closely variations in predictions mirror variations in human ratings. **Spearman Rank Correlation Coefficient (SRCC)**: SRCC Sedgwick (2014) evaluates the consistency in rank ordering between predictions and ground truth, regardless of absolute values. It is especially useful when relative ranking is more important than exact numeric scores. **Kendall's Tau Rank Correlation (KTAU)**: KTAU McLeod (2005) is a rank-based measure that assesses the strength and direction of association between predicted and actual rankings. Compared to SRCC, it is more robust to ties and small rank differences, providing complementary insights into ranking performance.

---

[2] https://github.com/tencent-ailab/MuQ

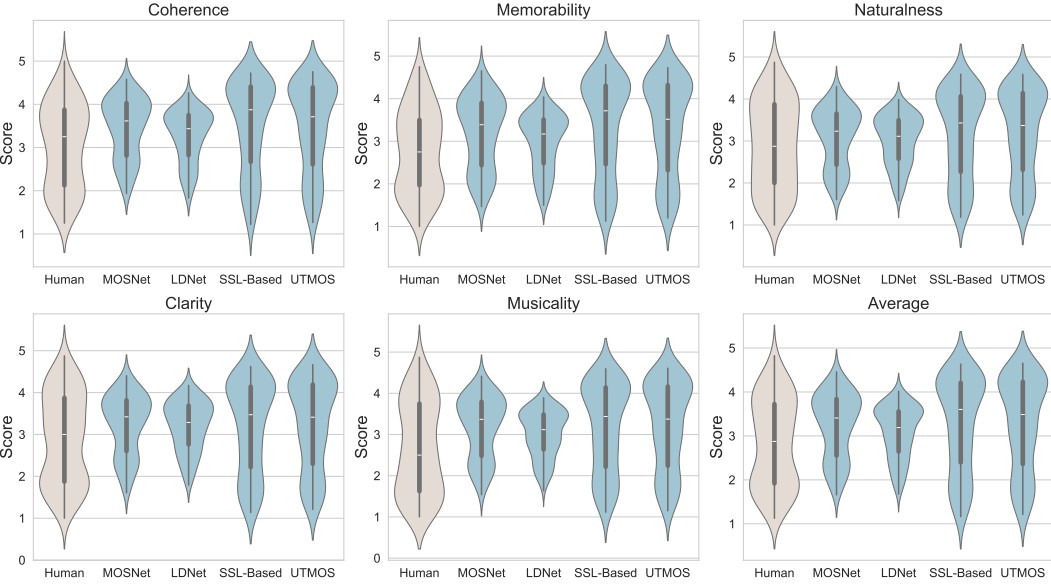

Figure 5: Violin plots of the aesthetic evaluation results between human annotation and different prediction systems.

## 5 EXPERIMENTAL RESULTS

### 5.1 AESTHETIC EVALUATION

To evaluate the effectiveness of SongEval, we train the assessment models on SongEval and conduct a comprehensive analysis of their performance across the five aesthetic dimensions. Following the best practices established in the VoiceMOS Challenge Huang et al. (2022a); Cooper et al. (2023), we evaluate results from two evaluation perspectives: 1) **utterance-level** evaluation directly compares model predictions with each individual human rating, offering a fine-grained measure of perceptual alignment and capturing subjective variability in annotation; 2) **system-level** evaluation first aggregates scores across all samples per model and then compares the predicted mean score with the corresponding human-annotated average, reflecting a more holistic view of each model's ability to assess overall musical quality.

The results across both evaluation levels are presented in Table 3. We can observe that all models trained on SongEval can reasonably predict multi-dimensional aesthetic scores, with SSL-based and UTMOS-based models demonstrating consistently superior performance across all five dimensions, particularly in coherence and structural clarity. This suggests that models benefiting from pretrained self-supervised features are better able to model high-level musical structure. To further illustrate the behavior of each system in modeling different aspects of musical aesthetics, we visualize the predicted scores and human-annotated scores across all five aesthetic dimensions using violin plots, as shown in Figure 5. These plots highlight the distribution between model-predicted scores and human-annotated scores. Across most dimensions, SSL-based and UTMOS-based models show tighter distributions and closer alignment with the annotated distribution, indicating their ability to replicate the full score spectrum observed in human ratings. In contrast, MOSNet-based and LDNet-based tend to produce more narrower or biased distributions. This suggests these systems may underfit or overly generalize these complex perceptual cues.

These results collectively demonstrate the effectiveness and robustness of the SongEval dataset as a training resource. Unlike prior datasets that are limited in genre diversity, song completeness, or annotation richness, SongEval enables systems to generalize across a wide range of aesthetic attributes, music styles, and languages. The fact that all evaluated systems achieve stable and interpretable scores across dimensions confirms that SongEval provides consistent, high-quality supervision for training reliable music aesthetic prediction models. These findings not only validate the design of SongEval but also underscore its unique contribution as the first open-source, large-scale dataset designed specifically for holistic song-level aesthetic evaluation.

## 5.2 COMPARISON WITH OTHER OBJECTIVE METRICS

To further validate the effectiveness of models trained on the SongEval dataset as an aesthetic evaluation metric, we compare their performance with several widely used objective metrics commonly employed for evaluating song generation. These metrics include: four perceived audio aesthetic metrics from Audiobox-Aesthetic Tjandra et al. (2025a), including Production Quality (PQ), Production Complexity (PC), Content Enjoyment (CE), and Content Usefulness (CU), song-level vocal range for measuring vocal agility, quantifying and flexibility Yuan et al. (2025).

Table 4: Pearson correlation between annotated aesthetic score and objective metrics. The results are compared in the musicality aspect and the average of all aspects.

|  | CE | CU | PC | PQ | Vocal Range | Aesthetic (Ours) |
|---|---|---|---|---|---|---|
| Coherence | 0.631 | 0.679 | 0.433 | 0.636 | 0.657 | 0.917 |
| Memorability | 0.605 | 0.654 | 0.400 | 0.625 | 0.667 | 0.910 |
| Naturalness | 0.602 | 0.645 | 0.396 | 0.616 | 0.739 | 0.909 |
| Clarity | 0.574 | 0.627 | 0.394 | 0.603 | 0.694 | 0.908 |
| Musicality | 0.608 | 0.653 | 0.388 | 0.622 | 0.751 | 0.916 |
| Average | 0.614 | 0.662 | 0.408 | 0.630 | 0.702 | 0.912 |

Each song is evaluated by both the prediction models trained on SongEval and the conventional objective metrics. We then compute the Pearson correlation Sedgwick (2012) between each metric's prediction and the human-annotated aesthetic scores. We employ the UTMOS-based system trained on SongEval as a representative system. The comparative results are shown in Table 4. The UTMOS-based system trained on the SongEval dataset consistently demonstrates stronger correlation with human aesthetic annotation across all five dimensions, particularly in coherence and musicality, which are more semantically driven and less captured by low-level acoustic measures.

Among AudioBox metrics, PC shows significantly lower correlation across all five aesthetic dimensions, while other metrics perform relatively better in evaluating musicality and structural clarity. Additionally, Vocal Range proves effective in detecting the presence of singing but lacks sensitivity to more nuanced aspects such as memorability and phrasing naturalness. In contrast, aesthetic evaluation models trained on the SongEval dataset consistently achieve higher alignment with human ratings across all five proposed dimensions. This demonstrates the necessity of a dedicated, perceptually grounded dataset like SongEval to enable holistic and meaningful evaluation of generative song systems. Rather than replacing traditional metrics, SongEval trained models complement them by addressing the aesthetic and experiential gaps left unfilled by existing approaches.

## 6 CONCLUSION

In this study, we present SongEval, the first benchmark dataset dedicated to musical aesthetics evaluation. The dataset contains 2,399 full-length songs totaling over 140 hours, annotated by 16 professional annotators across five carefully defined aesthetic dimensions: overall coherence, memorability, vocal phrasing naturalness, structural clarity, and musicality. The songs span both English and Chinese languages and cover nine common musical genres, ensuring linguistic and stylistic diversity. All annotations are rated on a 1–5 scale and are based on rigorous guidelines to ensure consistency and reliability. Experimental results demonstrate that models trained on the SongEval outperform existing objective audio metrics in predicting human-perceived musical aesthetics. We expect SongEval to serve as a strong foundation for future work in controllable music generation, quality assessment, and style transfer.

For future work, our primary goal is to develop more robust and fine-grained tools for automatic aesthetic evaluation based on the proposed SongEval dataset. Design advanced predictive models that better capture subjective aesthetic signals and generalize across musical styles, genres, and languages.

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

## A  APPENDIX

In the Appendix:

- A.1 We discuss the limitations of this study and future work.
- A.2 We provide the characteristic details of the song generated by different systems.
- A.3. We describe the filter process of generated songs.
- A.4. We provide the definitions of each point on the 1-5 aesthetics score.
- A.5. We provide details of the annotation process.

### A.1  LIMITATIONS AND FUTURE WORK

While the SongEval dataset establishes a foundation for song aesthetic evaluation, some limitations warrant further exploration. One key limitation lies in the potential correlation and overlap among the five defined aesthetic dimensions. For instance, dimensions such as overall coherence and structural clarity, or musicality and memorability, may exhibit high interdependence in practice. This is partially due to the holistic nature of song perception, where multiple musical aspects often influence a listener's judgment simultaneously. Despite our efforts to provide clear annotation guidelines and expert training to distinguish these dimensions, subjective perception inherently involves cognitive and emotional entanglement, making absolute separation between factors challenging. Nevertheless, we argue that this limitation reflects a psychologically grounded view of how listeners experience song and does not undermine the value of the dataset.

### A.2  CHARACTERISTIC DETAILS OF GENERATED SONGS

To ensure consistency across samples from different music generation systems, we standardized and documented the audio format for all generated songs. Table 5 summarizes the sampling rate and channel configuration used by each system. YUE Yuan et al. (2025) produces mono-channel audio; to maintain uniformity for downstream processing and model training, we duplicated the mono channel to simulate a stereo channel signal.

Table 5: Characteristic details of generated songs over different systems.

|  | Suno | Udio | Mureka | YUE | DiffRhythm |
|---|---|---|---|---|---|
| Sampling Rate | 48000 | 48000 | 44100 | 44100 | 44100 |
| Channel | 2 | 2 | 2 | 1 | 2 |

### A.3 FILTER PROCESS OF GENERATED SONGS

We employ vocal range to analyze the vocal components of each generated song to detect cases lacking singing voice, such as instrumental-only tracks or speech-like readings. Samples that do not meet the minimum vocal characteristics expected in a sung performance are excluded from the dataset. This ensures that all retained samples exhibit meaningful vocal content consistent with the intended song structure and aesthetic.

### A.4 DEFINITIONS OF EACH POINT ON THE 1-5 SCALE

We define a framework that focuses on five key dimensions, each assessed on a 1-5 point scale, where higher scores indicate greater proficiency and adherence to established musical principles. These dimensions aim to provide a detailed and objective measure of a piece's overall quality, from its structural integrity to its emotional impact and memorability. The scoring criteria for each dimension are defined below, offering clear benchmarks for evaluators.

- Coherence: This dimension assesses the interconnectedness and emotional unity across various musical sections (e.g., introduction, verse, chorus, bridge, outro) within a piece. It evaluates the natural flow and dynamic integration between these elements.

- Memorability: This dimension evaluates whether the piece contains elements that are easily remembered or recognized after a single listening, such as distinctive melodic hooks, rhythmic patterns, or unique instrumental motifs.

- Naturalness: This dimension evaluates the appropriateness of breath control, articulation, and the alignment of vocal delivery with the lyrical meaning, rhythmic structure, and melodic flow of the song. Inappropriate vocal delivery can significantly disrupt the listening experience.

- Clarity: This dimension assesses the clarity and logical arrangement of the song's structural sections (e.g., intro, verse, chorus, bridge, outro). It also considers whether the arrangement follows traditional song structures or presents a clear and effective innovative design.

- Musicality: This dimension assesses the overall sensory comfort and pleasantness of the musical experience, considering the mix, balance, timbre, arrangement, vocal performance, and the synergy between different musical elements (e.g., human voice and instrumental accompaniment).

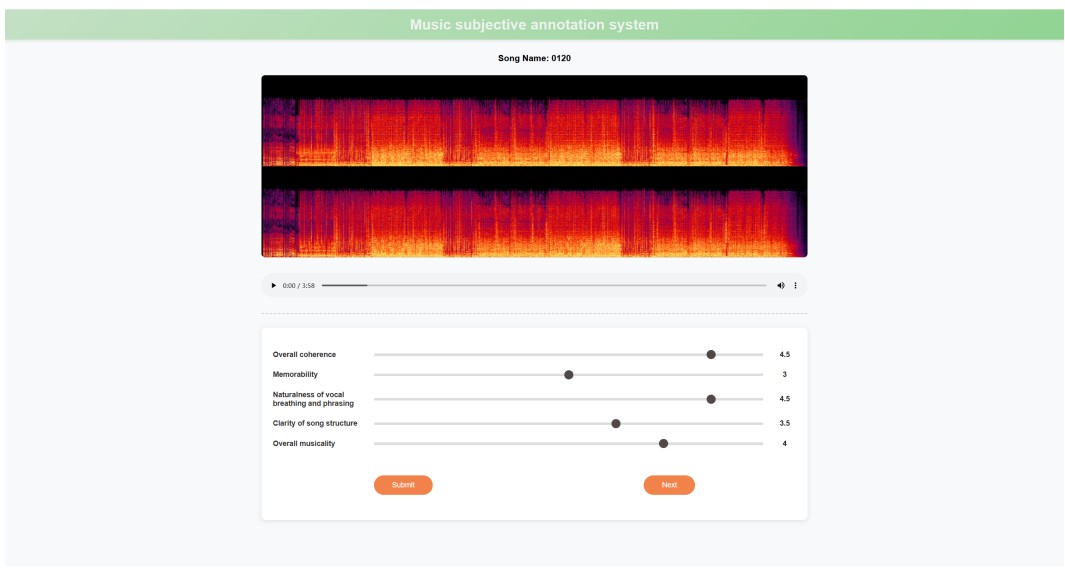

Figure 6: Screenshot of subjective annotation interface used for evaluating musical aesthetics.

## A.5 Details of annotation process

To ensure reliable and standardized subjective annotation, we employed a web-based annotation platform (as shown in Figure 6) that integrates both audio playback and visual spectrogram display. Annotators were asked to listen to the full song before assigning ratings across five aesthetic dimensions using intuitive sliders, each ranging from 1 (very poor) to 5 (excellent). The interface was designed for clarity and efficiency, facilitating streamlined submission and navigation between songs.

To guarantee high-quality and unbiased annotations, we collaborated with an independent third-party team specializing in audio annotation. This team was responsible for managing the annotation workflow, verifying annotator qualifications, and monitoring consistency throughout the process. Annotators were selected based on their musical background or relevant auditory experience, and were given detailed training on the five aesthetic criteria.

Each annotator was compensated at a rate of $5 USD per song, calibrated to reflect the average song duration (2–6 minutes) and required attention. In total, annotations were collected for 2,399 songs, with the complete annotation process managed and quality-controlled by the third-party team. On the generation side, to build a musically diverse and high-quality dataset, we accessed three commercial song generation systems—Udio, Suno, and Mureka—through official APIs or premium memberships. These services required monthly subscriptions or credit-based payments, averaging $30 USD per system. The total cost for song generation and access rights amounted to approximately $48,000 USD, including necessary premium plans for exporting full-length tracks.

