# OpenReview forum: "SongEval: A Benchmark Dataset for Song Aesthetics Evaluation"
_ICLR.cc/2026/Conference — ICLR 2026 Conference Withdrawn Submission_

### Official Review · Reviewer_hYm1 · 2025-10-31

**Soundness:** 2
**Presentation:** 2
**Contribution:** 2
**Rating:** 2
**Confidence:** 3

**Summary:**

SongEval is a large-scale benchmark dataset designed to evaluate the aesthetic quality of full-length AI-generated songs. Recognizing that existing objective metrics fail to capture subjective human perception, the dataset includes 2,399 English and Chinese songs (over 140 hours) spanning nine musical genres. Its core contribution lies in the detailed human annotation: 16 professional musicians evaluated each piece on a 1–5 scale across five aesthetic dimensions—overall coherence, memorability, naturalness of vocal phrasing and breathing, clarity of song structure, and overall musicality. Experimental results show that models trained with SongEval correlate more strongly with human judgments and significantly outperform traditional objective metrics in predicting perceived musical quality.

**Strengths:**

-The paper introduces SongEval, a large-scale, open-source benchmark dataset that addresses an important and challenging problem: evaluating the aesthetic quality of full-length AI-generated songs.
-The dataset is both substantial and diverse, comprising 2,399 songs (over 140 hours of audio) in English and Chinese across nine musical genres.
-The annotations are provided by 16 professional musicians, offering high-quality expert evaluations that strengthen the reliability and credibility of the benchmark.
-The dataset employs a multi-dimensional 1–5 rating scale across five well-defined aesthetic aspects, rather than relying on a single aggregate score, enabling more nuanced assessment.
-The paper demonstrates through experiments that models trained on SongEval correlate more strongly with human perception and significantly outperform traditional objective metrics.

**Weaknesses:**

-The methodological basis for the chosen subjective evaluation dimensions is not well justified. The paper does not demonstrate evidence of a systematic literature review or structured expert interviews to support the selection of the five aesthetic criteria.
-The overall musicality dimension is broad and conceptually ambiguous. Even with the provided definition, its interpretive scope risks introducing significant variance among annotators, especially since it aggregates multiple perceptual qualities into a single score.
-The authors acknowledge in Appendix A.1 that these aesthetic factors are difficult to disentangle due to “cognitive and emotional entanglement,” which underscores a foundational ambiguity in the benchmark’s conceptual framework.
-The presentation of experimental results lacks statistical rigor. For instance, Table 3 reports mean values without standard deviations, preventing assessment of variance or reliability in the comparisons.
-The high correlation of the “Aesthetic (Ours)” score in Table 4 is potentially misleading, as this metric is derived from a UTMOS-based system trained directly on SongEval’s own subjective labels. This constitutes a form of circular evaluation and does not provide meaningful comparative insight.
-The set of “objective metrics” used for baseline comparison appears insufficient, as they do not reflect fundamental elements of musical composition or perceptual music theory, limiting the interpretive value of the comparison.
-The correlation analysis in Table 4 is overly aggregated. A more informative analysis would examine correlations across different genres or stylistic categories to determine whether the benchmark generalizes beyond broad average trends.
-The paper does not provide background details regarding the annotators’ training, specialization, or experience, which limits the reader’s ability to assess the reliability and consistency of the expert ratings.

**Questions:**

-	Beyond the number of annotated dimensions, what qualitative characteristics of SongEval make it more reliable or representative for evaluating musical aesthetics than existing benchmarks?
-	Since the dataset includes songs generated by five different models, did the authors observe any statistically meaningful differences in human aesthetic ratings across these model types?
-	What annotation guidelines or protocols were provided to the evaluators? Is there any reported measure of inter-rater agreement (e.g., ICC, Cohen’s κ)?
-	Given the acknowledged conceptual overlap among the five aesthetic dimensions, have the authors conducted correlation analysis or factor analysis to verify that each dimension captures a distinct perceptual construct?
-	Were any linguistic or cultural effects observed in ratings between English and Chinese songs?

---

### Official Review · Reviewer_EXKt · 2025-10-31

**Soundness:** 4
**Presentation:** 4
**Contribution:** 3
**Rating:** 6
**Confidence:** 5

**Summary:**

This paper presents SongEval, the first open-source, large-scale benchmark dataset dedicated to evaluating the aesthetics of full-length songs, addressing the long-standing challenge of assessing the subjective aesthetic quality of generated songs. The dataset comprises 2,399 full-length songs, totaling over 140 hours of audio, covering both English and Chinese languages and nine mainstream music genres. Aesthetic annotations are provided by 16 professional annotators with musical backgrounds, who evaluate each song across five key dimensions: overall coherence, memorability, naturalness of vocal breathing and phrasing, clarity of song structure, and overall musicality.

The paper concludes that SongEval fills a critical gap in existing music evaluation datasets and provides a reliable foundation for future research in controllable music generation, quality assessment, and style transfer. The dataset and associated toolkit are made publicly available to facilitate further advancements in the field.

**Strengths:**

With 2,399 full-length songs (140+ hours), SongEval is significantly larger than existing alternatives. It covers two languages (English, Chinese) and nine genres, and includes both vocals and accompaniment—addressing the limitation of single-component focus in prior datasets.
The authors test four models with distinct architectures (convolutional, self-supervised, ensemble-based) and evaluate performance at both utterance and system levels. Direct comparisons to established objective metrics (e.g., Audiobox-Aesthetic’s PC, Vocal Range) clearly demonstrate SongEval’s superiority in aligning with human aesthetic perception.

**Weaknesses:**

The dataset is dominated by generated songs from five models, with only a small subset of non-copyrighted real songs. This narrow focus on generated content limits its ability to evaluate models on real-world, human-composed music—a critical use case for aesthetic evaluation.

**Questions:**

Could the author provide quantitative measures of inter-annotator agreement (e.g., Krippendorff's alpha, Cohen's kappa) for each of the five aesthetic dimensions? This would help confirm the reliability of the annotation process.

---

### Official Review · Reviewer_5n43 · 2025-11-01

**Soundness:** 2
**Presentation:** 3
**Contribution:** 2
**Rating:** 2
**Confidence:** 3

**Summary:**

This paper proposes a new benchmark dataset named SongEval to assess the aesthetics of generated songs. This benchmark utilizes several existing song generation models such as DiffRhythm to generate songs with different genres, incorporating lyrics in English and Chineses, and then exploits four human experts to annotate the generated songs with five categories. This newly introduced dataset is largest in terms of size and annotation categories compared to prior datasets. To demonstrate the effectiveness of the dataset and song aesthetic evaluation, the authors train four models to predict aesthetic scores and demonstrate better performance than existing objective evaluation metrics in predicting human-perceived musical quality.

**Strengths:**

- It is a relevant problem to study and assess song aesthetics, and the newly introduce benchmark dataset may be useful to the community.
- In experiments, it was shown that the trained models to predict aesthetic scores achieves better performance than existing objective evaluation metrics in predicting human-perceived musical quality.
- Writing is easy to follow.

**Weaknesses:**

- The authors use song generation models to generate songs rather than actual songs. So the qualities of them largely depend on the song generation model and it may be desirable to use actual songs for building the benchmark and annotations.
- My main concern is the experiment designs to show the effectiveness of the new dataset. This paper first proposes to compare the predictions from the trained evaluation models on the dataset, with the human annotations. Even though the reported results look good, there is no comparison to show the effectiveness of the dataset rather than the prediction models. I think the more relevant experiment would be training song generation model with the annotations as human preference alignment. Furthermore, in the comparison with objective evaluation metrics, it is not surprising that the trained model on the dataset has higher correlation since such objective metrics may capture different aspects and the trained model is directly trained on the same categories' annotations.
- The annotation qualities are also questionable. In Figure 3, it was shown that almost half of the annotations have the highest scores (5). So the annotation instruction (and/or human expert qualification) may be problematic.

**Questions:**

- As shown in Figure 4 right, why are there uneven uses of different song generation models, e.g., DiffRhythm is used more than others?

Please see and address the weaknesses above.

---

### Official Review · Reviewer_WNQ2 · 2025-11-03

**Soundness:** 2
**Presentation:** 3
**Contribution:** 2
**Rating:** 4
**Confidence:** 2

**Summary:**

Aesthetics serve as an implicit and important criterion in song generation tasks that reflect human perception beyond objective metrics. However, evaluating the aesthetics of generated songs remains a fundamental challenge, as the appreciation of music is highly subjective. Existing evaluation metrics, such as embedding-based distances, are limited in reflecting the subjective and perceptual aspects that define musical appeal. To address this issue, we introduce SongEval, the first open-source, large-scale benchmark dataset for evaluating the aesthetics of full-length songs. SongEval includes over 2,399 songs in full length, summing up to more than 140 hours, with aesthetic ratings from 16 professional annotators with musical backgrounds. Each song is evaluated across five key dimensions: overall coherence, memorability, naturalness of vocal breathing and phrasing, clarity of song structure, and overall musicality. The dataset covers both English and Chinese songs, spanning nine mainstream genres. Moreover, to assess the effectiveness of song aesthetic evaluation, we conduct experiments using SongEval to predict aesthetic scores and demonstrate better performance than existing objective evaluation metrics in predicting human-perceived musical quality.

**Strengths:**

Comprehensive Dataset: SongEval provides a large-scale benchmark dataset with over 2,399 full-length songs, allowing for extensive analysis and evaluation of song aesthetics across various genres and languages.

Multi-Dimensional Evaluation: The dataset includes aesthetic ratings across five key dimensions (overall coherence, memorability, naturalness, clarity, and musicality), offering a nuanced approach to assessing musical appeal beyond simple metrics.

Improved Predictive Performance: Experiments demonstrate that SongEval outperforms existing objective evaluation metrics in predicting human-perceived musical quality, highlighting its effectiveness in evaluating song aesthetics.

**Weaknesses:**

Subjectivity of Aesthetic Evaluation: The appreciation of music is inherently subjective, and even with professional annotators, individual biases may influence the aesthetic ratings, potentially affecting the dataset's reliability.

Limited Scope of Genres: While SongEval covers nine mainstream genres, it may not encompass all musical styles, which could limit its applicability for evaluating songs from less represented genres or niche markets.

Dependence on Annotator Expertise: The quality of the aesthetic ratings relies heavily on the expertise of the 16 professional annotators, and any inconsistencies in their evaluations could impact the overall validity of the dataset.

**Questions:**

1) How can minimize individual biases among annotators to ensure more consistent and reliable aesthetic ratings in the SongEval dataset?

2) What strategies can be employed to expand the dataset to include a wider variety of musical genres, particularly those that are underrepresented in the current collection?

3) How can we ensure that the evaluations remain valid and reliable if the expertise of the annotators varies, and what measures can be taken to standardize their assessments?

---

### Note · Authors · 2025-11-24

I have read and agree with the venue's withdrawal policy on behalf of myself and my co-authors.